# COMPOUND DENSITY NETWORKS

## ABSTRACT

Despite the huge success of deep neural networks (NNs), finding good mechanisms for quantifying their prediction uncertainty is still an open problem. It was recently shown, that using an ensemble of NNs trained with a proper scoring rule leads to results competitive to those of Bayesian NNs. This ensemble method can be understood as finite mixture model with uniform mixing weights. We build on this mixture model approach and increase its flexibility by replacing the fixed mixing weights by an adaptive, input-dependent distribution (specifying the probability of each component) represented by an NN, and by considering uncountably many mixture components. The resulting model can be seen as the continuous counterpart to mixture density networks and is therefore referred to as *compound density networks*. We empirically show that the proposed model results in better uncertainty estimates and is more robust to adversarial examples than previous approaches.

## 1 INTRODUCTION

Deep neural networks (NNs) have achieved state-of-the-art performance in many application areas, such as computer vision (Krizhevsky et al., 2012) and natural language processing (Collobert et al., 2011). However, despite achieving impressive prediction accuracy on these supervised machine learning tasks, NNs do not provide good ways of quantifying predictive uncertainty. This is undesirable for many mission critical applications, where taking wrong predictions with high confidence could have fatal consequences (e.g. in medical diagnostics or autonomous driving).

A principled and the most explored way to quantify the uncertainty in NNs is through Bayesian inference. In the so-called Bayesian neural networks (BNNs) (Neal, 1995), the NN parameters are treated as random variables and the goal of learning is to infer the posterior probability distribution of the parameters given the training data. Since exact Bayesian inference in NNs is computationally intractable, different approximation techniques have been proposed (Neal, 1995; Blundell et al., 2015; Hernández-Lobato & Adams, 2015; Ritter et al., 2018, etc.). Given the (approximate) posterior, the final predictive distribution is obtained as the expected distribution under the posterior, where the expectation can be seen as an ensemble of an uncountably infinite number of predictors, where the prediction of each model is weighted by the posterior probability of the corresponding parameters.

Based on a Bayesian interpretation of dropout (Srivastava et al., 2014), Gal & Ghahramani (2016) proposed to apply it not only during training but also when making predictions to estimate predictive uncertainty. Interestingly, dropout has been also interpreted as ensemble model (Srivastava et al., 2014) where the predictions are averaged over the different NNs resulting from different dropout-masks. Inspired by this, Lakshminarayanan et al. (2017) proposed to use an simple NN ensemble to quantify the prediction uncertainty, i.e. to train a set of independent NNs using a proper scoring rule and defining the final prediction as the arithmetic mean of the outputs of the individual models, which corresponds to defining a uniformly-weighted mixture model. It is argued, that the model is able to encode two sources of uncertainty by calibrating the prediction uncertainty in each component and capturing the "model uncertainty" by averaging over the components. Note, that this should hold for any kind of predictive model that is defined in terms of a mixture distribution.

In this paper, we therefore aim at further investigating the potential that lies in using mixture distributions for uncertainty quantification. The flexibility of a mixture model can be increased by learning input-conditioned mixture weights, like it is done by mixture density networks (MDN) (Bishop,

1994). Furthermore, one can consider uncountably many component distributions instead of a finite set, which turns the mixture distribution into a compound distribution. We combine both by deriving the continuous counterpart of MDNs, which we call *compound density networks* (CDNs). This model corresponds to a compound distribution in which both, component and mixing distribution, are parametrized based on NNs. We experimentally show that CDNs allow for better uncertainty quantification and are more robust to adversarial examples than previous approaches.

This paper is organized as follows. In Section 2 we give a brief introduction to MDNs. We then formally define CDNs in Section 3. We review related work in Section 5 and present a detailed experimental analysis in Section 6. Finally we conclude our paper in Section 7.

## 2 MIXTURE DENSITY NETWORKS

Let $\mathcal{D} = \{\mathbf{x}_n, \mathbf{y}_n\}_{n=1}^N$ be a i.i.d dataset and let us define the following conditional mixture model

$$p(\mathbf{y}|\mathbf{x}) = \sum_{k=1}^K p\left(\mathbf{y}|\boldsymbol{\phi}_k(\mathbf{x})\right) p(\boldsymbol{\phi}_k(\mathbf{x})|\boldsymbol{\pi}(\mathbf{x})) \ , \tag{1}$$

and an NN that maps $\mathbf{x}$ onto both the parameters $\boldsymbol{\pi}(\mathbf{x})$ of the mixing distribution and the parameters $\{\boldsymbol{\phi}_k(\mathbf{x})\}_{k=1}^K$ of the $K$ mixture components. The complete system is called mixture density network (MDN) and was proposed by Bishop (1994). That is, an MDN is an NN parametrizing a conditional mixture distribution, where both the mixture components and the mixture coefficients depend on input $\mathbf{x}$.[1] MDNs can be trained by maximizing the log-likelihood of the parameters of the NN given the training set $\mathcal{D}$ using gradient-based optimizers such as stochastic gradient descent (SGD) and its variants.

MDNs belong to a broader class of models called mixture of experts (MoE) (Jacobs et al., 1991) which differ from standard mixture models by assuming that the mixture coefficients depend on the input.[2] Because of its formulation as a mixture distribution, the predictive distribution of an MDN can handle multimodality better than a standard discriminative neural network. Furthermore, the mixture coefficients allow us to encode uncertainty about the prediction, namely by modelling the probability from which component distribution a data point was sampled.

## 3 COMPOUND DENSITY NETWORKS

We aim at generalizing the MDN from a finite mixture distribution to a mixture of an uncountable set of components. The continuous counterpart of a conditional mixture distribution in eq. (1) is given by the conditional *compound probability distribution*

$$p(\mathbf{y}|\mathbf{x}) = \int p(\mathbf{y}|\boldsymbol{\phi}(\mathbf{x}))p(\boldsymbol{\phi}(\mathbf{x})|\boldsymbol{\pi}(\mathbf{x})) \, \mathrm{d}\boldsymbol{\phi}(\mathbf{x}) \ , \tag{2}$$

where $\boldsymbol{\phi}(\mathbf{x})$ turns from a discrete into a continuous random variable.

We now want to follow the approach of MDNs to model the parameters of the components and the mixing distribution by NNs. To handle the continuous state space of $\boldsymbol{\phi}(\mathbf{x})$ in the case of a compound distributions, the key idea is now to let $\boldsymbol{\phi}(\mathbf{x})$ be given by a stochastic NN $f(\mathbf{x}; \boldsymbol{\theta}) = \boldsymbol{\phi}(\mathbf{x})$ with stochastic parameters $\boldsymbol{\theta}$. Since given $\mathbf{x}$, $f$ is a deterministic map from $\boldsymbol{\theta}$ to $\boldsymbol{\phi}(\mathbf{x})$, it is possible to replace the mixing distribution $p(\boldsymbol{\phi}(\mathbf{x})|\boldsymbol{\pi}(\mathbf{x})) = p(f(\mathbf{x}; \boldsymbol{\theta})|\boldsymbol{\pi}(\mathbf{x}))$ by a distribution $p(\boldsymbol{\theta}|\boldsymbol{\pi}(\mathbf{x}))$ over $\boldsymbol{\theta}$. We further assume, that the parameters $\boldsymbol{\pi}(\mathbf{x})$ of the mixing distribution are given by some parametrized function $g(\mathbf{x}; \boldsymbol{\psi}) = \boldsymbol{\pi}(\mathbf{x})$ which can also be modelled based on NNs. In correspondence to MDNs, the complete system is called *compound density network* (CDN) and it is summarized by the following equation

$$p(\mathbf{y}|\mathbf{x}; \boldsymbol{\psi}) = \int p(\mathbf{y}|f(\mathbf{x}; \boldsymbol{\theta}))p(\boldsymbol{\theta}|g(\mathbf{x}; \boldsymbol{\psi})) \, \mathrm{d}\boldsymbol{\theta} = \mathbb{E}_{p(\boldsymbol{\theta}|g(\mathbf{x}; \boldsymbol{\psi}))}[p(\mathbf{y}|f(\mathbf{x}; \boldsymbol{\theta}))] \ . \tag{3}$$

---

[1]For instance, as in the original publication, the mixture components could be $K$ Gaussians, with $\boldsymbol{\phi}_k(\mathbf{x})$ being input-specific means and variances, and the mixture probabilities could be given by (applying the softmax function to the unnormalized) mixing weights $\boldsymbol{\pi}(\mathbf{x})$, both computed by one NN.

[2]See (Bishop, 2006, ch. 5.6 and ch. 14.5.3) and (Murphy, 2012, ch. 11.2.4) for a detailed discussion of MDNs.

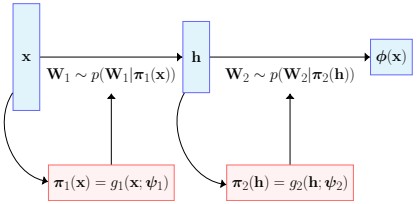

Figure 1: An example of a probabilistic hypernetwork applied to a two-layer MLP.

As for MDNs, CDNs training corresponds to maximizing its likelihood function, which is given by

$$\log p(\mathcal{D}|\boldsymbol{\psi}) = \sum_{n=1}^{N} \log \mathbb{E}_{p(\boldsymbol{\theta}|g(\mathbf{x}_n;\boldsymbol{\psi}))} [\, p(\mathbf{y}_n|f(\mathbf{x}_n;\boldsymbol{\theta})] \ . \tag{4}$$

As we now have to deal with an integral instead of a finite sum, this log-likelihood function is intractable. Fortunately, applying the reparametrization trick (Kingma & Welling, 2014) to $\boldsymbol{\theta} \sim p(\boldsymbol{\theta}|g(\mathbf{x};\boldsymbol{\psi}))$ enables Monte Carlo integration, while still being able to optimize the log-likelihood via backpropagation in conjunction with SGD.

To avoid overfitting, we can regularize the mixing distribution $p(\boldsymbol{\theta}|g(\mathbf{x};\boldsymbol{\psi}))$ by penalizing its KL-divergence w.r.t. some simple distribution $p(\boldsymbol{\theta})$[3], leading to the regularized objective function

$$\mathcal{L}(\boldsymbol{\psi}) = \log p(\mathcal{D}|\boldsymbol{\psi}) - \lambda \sum_{n=1}^{N} D_{\mathrm{KL}}[\, p(\boldsymbol{\theta}|g(\mathbf{x}_n;\boldsymbol{\psi}))\|p(\boldsymbol{\theta})] \ , \tag{5}$$

where $\lambda$ controls the regularization strength. We summarize the training procedure of CDNs in the pseudo-code provided in Appendix A. Note, that CDNs correspond to an abstract framework for modelling compound distributions with NNs. In the following section, we present a concrete example on how CDNs can be implemented.[4]

### 3.1 PROBABILISTIC HYPERNETWORKS

Ha et al. (2017) proposed to (deterministically) generate the parameters of an NN by another NN, which they call the *hypernetwork*.[5] We would like to follow this approach for modelling a CDN, that is, we aim at modelling the mixing distribution $p(\boldsymbol{\theta}|g(\mathbf{x};\boldsymbol{\psi}))$ over network parameters by NNs. Since now the hypernetworks maps $\mathbf{x}$ to a distribution over parameters instead of a specific value $\boldsymbol{\theta}$, we refer to them as *probabilistic hypernetworks*. In the following we will describe this idea in more detail.

Let $f$ be a multi-layer perceptron[6] (MLP) with $L$-layers, parametrized by a set of layers' weight matrices[7] $\boldsymbol{\theta} = \{\mathbf{W}_l\}_{l=1}^{L}$, that computes the parameters $\boldsymbol{\phi}(\mathbf{x}) = f(\mathbf{x};\boldsymbol{\theta})$ of the CDNs component distribution in eq. (3). Let $\mathbf{h}_1, \ldots, \mathbf{h}_{l-1}$ denote the states of the hidden layers, and let us define $\mathbf{h}_0 = \mathbf{x}$, and $\mathbf{h}_L = f(\mathbf{x};\boldsymbol{\theta})$. We now assume the weight matrices $\{\mathbf{W}_l\}_{l=1}^{L}$ to be random variables and to be independent from each other given the state of the previous hidden layer. We define a series of probabilistic hypernetworks $g = \{g_l\}_{l=1}^{L}$ (parametrized by $\boldsymbol{\psi} = \{\boldsymbol{\psi}_l\}_{l=1}^{L}$), where $g_l$ maps $\mathbf{h}_{l-1}$ to the parameters of the distribution of $\mathbf{W}_l$, and let the joint distribution over $\boldsymbol{\theta}$ be given by

$$p(\boldsymbol{\theta}|g(\mathbf{x};\boldsymbol{\psi})) = \prod_{l=1}^{L} p(\mathbf{W}_l|g_l(\mathbf{h}_{l-1};\boldsymbol{\psi}_l)) \ . \tag{6}$$

An illustration of a stochastic two-layer network $f(\mathbf{x};\boldsymbol{\theta})$ computing $\boldsymbol{\phi}(\mathbf{x})$, with parameters distributions given by probabilistic hypernetworks as defined in eq. (6) is given in Figure 1. To make

---

[3]Note, that another option is $L^p$ regularization on $\boldsymbol{\psi}$.

[4]An alternative implementation corresponding to a form of adaptive dropout is described in Appendix B.

[5]Specifically, they propose to apply a hypernetwork to compute the weight matrix of a recurrent NN at each time-step, given the current input and the previous hidden state.

[6]Note, that probabilistic hypernetworks can analogously be applied to model the distribution over parameters of any other kind of network.

[7]We assume that the bias parameters are absorbed into the weight matrix.

the definition of our model complete, we need to define the concrete statistical model we pick for $p(\mathbf{W}_l|g_l(\mathbf{h}_{l-1}; \boldsymbol{\psi}))$. This is done in the following section.

## 3.2 PROBABILISTIC HYPERNETWORKS WITH MATRIX VARIATE NORMALS

A statistical model that was recently applied as the posterior over weight matrices in BNNs (Louizos & Welling, 2016; Sun et al., 2017; Zhang et al., 2018; Ritter et al., 2018) is the matrix variate normal (MVN) distribution (Gupta & Nagar, 1999). An MVN is parametrized by three matrices: a mean matrix $\mathbf{M}$ and two covariance factor matrices $\mathbf{A}$ and $\mathbf{B}$. It is connected to the multivariate Gaussian by the following equivalence

$$\mathbf{X} \sim \mathcal{MN}(\mathbf{X}; \mathbf{M}, \mathbf{A}, \mathbf{B}) \iff \text{vec}(\mathbf{X}) \sim \mathcal{N}(\text{vec}(\mathbf{X}); \text{vec}(\mathbf{M}), \mathbf{A} \otimes \mathbf{B}) \ , \qquad (7)$$

where $\text{vec}(\mathbf{X})$ denotes the vectorization of $\mathbf{X}$. Due to the Kronecker factorization of the covariance, an MVN requires less parameters compared to a multivariate Gaussian, which motivates us to use it as the distribution over weight matrices in this work. Furthermore, we assume that the covariance factor matrices are diagonal matrices, following Louizos & Welling (2016). That is, we choose the mixture distribution of the CDN to be

$$p(\boldsymbol{\theta}|g(\mathbf{x}; \boldsymbol{\psi})) = \prod_{l=1}^{L} \mathcal{MN}(\mathbf{W}_l|g_l(\mathbf{h}_{l-1}; \boldsymbol{\psi}_l)) = \prod_{l=1}^{L} \mathcal{MN}(\mathbf{W}_l|\mathbf{M}_l, \text{diag}(\mathbf{a}_l), \text{diag}(\mathbf{b}_l)) \ , \qquad (8)$$

where $g_l$ maps the state $\mathbf{h}_{l-1}$ of the previous hidden layer onto the $l$-th MVN's parameters $\{\mathbf{M}_l, \mathbf{a}_l, \mathbf{b}_l\}$, defining the distribution over $\mathbf{W}_l$. Suppose $\mathbf{W}_l \in \mathbb{R}^{r \times c}$, then the corresponding MVN distribution has $rc + r + c$ parameters, which is more efficient compared to $rc + rc$ parameters when modeling $\mathbf{W}_l$ as fully-factorized Gaussian random variable.

In practice, the procedure of sampling $\boldsymbol{\theta}$ and computing $f(\mathbf{x}; \boldsymbol{\theta}) = \mathbf{h}_L$ now can be described by

$$\mathbf{h}_l = \sigma(\mathbf{h}_{l-1}\mathbf{W}_l) \ , \text{ where } \mathbf{W}_l \sim \mathcal{MN}(\mathbf{W}_l|g_l(\mathbf{h}_{l-l}; \boldsymbol{\psi}_l)) \ , \qquad (9)$$

for $l = 1, \dots, L$, where $\sigma$ is an arbitrary point-wise nonlinear activation function, which may differ for hidden and output layers.

Note that using the mixing distribution defined in eq. (8) allows us to apply the reparametrization trick (Appendix C.1). For the KL-divergence-based regularization, a straight forward choice for the simple distribution is $p(\boldsymbol{\theta}) = \prod_{l=1}^{L} \mathcal{MN}(\mathbf{W}_l|\mathbf{0}, \mathbf{I}, \mathbf{I})$, which corresponds to assuming that each layer's weight matrix is standard matrix normal distributed. In this case, the KL-term can be computed in closed form, as noted by Louizos & Welling (2016) and shown in Appendix C.2. Finally, we use a vector-scaling parametrization similar to the one used by Ha et al. (2017) and Krueger et al. (2017) for the mean matrices $\{\mathbf{M}_l\}_{l=1}^{L}$, which we explain in detail in Appendix D.

# 4 CONNECTION TO BAYESIAN NEURAL NETWORKS AND VARIATIONAL INFORMATION BOTTLENECK

Interestingly, eq. (2) could also be interpreted as integrating over the parameters of a Bayesian neural network, where $p(\boldsymbol{\phi}(\mathbf{x})|\boldsymbol{\pi}(\mathbf{x}))$ is an amortized approximate posterior. Performing variational inference in such a setting would lead to the following objective [8]

$$\text{ELBO}(\boldsymbol{\psi}) = \sum_{n=1}^{N} \mathbb{E}_{p(\boldsymbol{\theta}|g(\mathbf{x}_n; \boldsymbol{\psi}))} \big[ \log p(\mathbf{y}_n|f(\mathbf{x}_n; \boldsymbol{\theta})) - D_{\text{KL}}\big[ p(\boldsymbol{\theta}|g(\mathbf{x}_n; \boldsymbol{\psi}))\|p(\boldsymbol{\theta}) \big] \ . \qquad (10)$$

Note, that this is a lower bound of eq. (5) (by Jensen's inequality) and that stochastic approximations of both objectives become equivalent when they are based on a single sample of $\boldsymbol{\theta}$ and $\lambda = 1$. For arbitrary value of $\lambda$, the one-sample-approximation of eq. (5) is also equivalent to the approximated variational information bottleneck (VIB) objective (Alemi et al., 2017). Following this objective, the proposed approach corresponds to performing VIB where the network parameters, instead of the hidden units, are considered as latent variables. We will analyze the effects of following these different objectives experimentally in Section 6.4.

---

[8]Usually the approximate posterior does not depend on the input and therefore the KL term is independent from $\mathbf{x}$ as well. When performing mini-batch optimization in this case, it is common practice to re-scale the KL-term with $1/B$ where $B$ is the number of mini-batches as described by Graves (2011).

## 5 RELATED WORK

Various approaches for quantifying predictive uncertainty in NNs have been proposed. Applying Bayesian inference to NNs, i.e. treating the network parameters as random variables and estimating the posterior distribution given the training data based on Bayes' theorem, results in Bayesian neural networks (BNNs) (MacKay, 1992; Neal, 1995; Graves, 2011; Blundell et al., 2015; Louizos & Welling, 2016; Sun et al., 2017; Louizos & Welling, 2017; Ritter et al., 2018; Zhang et al., 2018, etc). Since the true posterior distribution is intractable, BNNs are trained based on approximate inference methods such as variational inference (VI) (Peterson, 1987; Hinton & Van Camp, 1993; Graves, 2011; Blundell et al., 2015), Markov Chain Monte Carlo (MCMC) (Neal, 1995), or Laplace approximation (MacKay, 1992; Ritter et al., 2018). The final prediction is then given by the expectation of the network prediction (given the parameters) w.r.t. the approximate posterior distribution. Louizos & Welling (2016) proposed to train BNNs based on VI with an MVN as the approximate posterior of each weight matrix (leading to a model they refer to as variational matrix Gaussian (VMG)). Multiplicative normalizing flow (MNF) (Louizos & Welling, 2017) models the approximate posterior as a compound distribution, where the mixing density is given by a normalizing flow. Zhang et al. (2018) also use an MVN approximate posterior and apply approximate natural gradient (Amari, 1998) based maximization on the VI objective, which results in an algorithm called noisy K-FAC. Meanwhile, the Kronecker-factored Laplace approximation (KFLA) (Ritter et al., 2018) extends the classical Laplace approximation by using an MVN approximate posterior with tractable and efficiently computed covariance factors, based on the Fisher information matrix.

There have been several concurrent works (Krueger et al., 2017; Louizos & Welling, 2017; Pawlowski et al., 2017; Sheikh et al., 2017) applying hypernetworks (Ha et al., 2017) to model the posterior distribution over network parameters in BNNs. In contrast to CDNs, the hypernetworks in these approaches are used to transform random noise drawn from a simple distribution into a random variable with a complicated distribution. That is, they use hypernetworks to sample from an implicit distribution of the parameters, without explicitly specifying the statistical model (like the MVN in our case). Krueger et al. (2017) and Louizos & Welling (2017) use normalizing flows, while Pawlowski et al. (2017) and Sheikh et al. (2017) use arbitrary NNs as their hypernetworks. The approach by Sheikh et al. (2017) is the one most closely related to ours, since they use an objective analogous to eq. (4). Note, that the main difference between CDNs and these hypernet-BNNs is that the approximate posterior in a Bayesian setting does not depend on the current input point, while the mixing distribution of the CDN does.

Gal & Ghahramani (2016) developed a theoretical framework that relates dropout training in NNs to approximate Bayesian inference and, as a result, proposed to approximate the predictive distribution by an average over the different networks resulting from independently sampled dropout-masks, a technique which they referred to as MC-dropout and which they applied to estimate the prediction uncertainty in NNs. Recently, Lakshminarayanan et al. (2017), proposed to use an ensemble of NNs in conjunction with a proper scoring rule and adversarial training to quantify the prediction uncertainty of deep NNs, leading to a model referred to as Deep Ensemble. The Deep Ensemble provides a non-Bayesian way to quantify prediction uncertainty, and is in this sense related to the approaches of Guo et al. (2017) and Hendrycks & Gimpel (2017).

## 6 EXPERIMENTS

We consider several standard tasks in our experimental analysis: 1D toy regression problems inspired by Hernández-Lobato & Adams (2015) and Depeweg et al. (2018) (Section 6.1), classification under out-of-distribution data (Section 6.2), and detection of and defense against adversarial examples (Szegedy et al., 2014) (Section 6.3). We consider the following recent models (described in Section 5) as the baselines for our CDN model: MNF, KFLA, noisy K-FAC, MC-dropout, and Deep Ensemble.[9]

In all of the experiments, we consider $\lambda \in \{10^{-n}\}_{n=1}^{5}$ when training CDNs based on eq. (5), and pick the highest value of $\lambda$ that still allows for high predictive power (e.g. an accuracy of $> 0.97$ on the validation set). We use this selection heuristic, based on our observation that in general, as $\lambda$

---

[9]We also investigated the VMG, an MoE, and an MDN. Due to space restrictions, results for these methods are reported in Appendix E.

increases, the accuracy is decreasing while the uncertainty estimate is increasing (see Appendix F.1 for an investigation of this behavior). The hyperparameters for the baselines are set to the values suggested by the respective original publications as outlined in Appendix F.

For BNNs and the CDN we estimate the predictive distribution $p(\mathbf{y}|\mathbf{x})$, based on 100 samples of network parameters from the posterior distribution or the mixing distribution, respectively. Unless stated otherwise, we use a single sample of $\boldsymbol{\theta}$ to do Monte Carlo integration during training. For MNIST and CIFAR-10 experiments, we use mini-batches of size 200. We use Adam (Kingma & Ba, 2015) with default hyperparameters for optimization in all experiments and the implementations provided by Louizos & Welling (2017)[10] and Zhang et al. (2018)[11] for MNF and noisy K-FAC, respectively. The implementation of all models and experiments will be made available to the public once the review process is concluded.

## 6.1 TOY REGRESSION

Following Hernández-Lobato & Adams (2015) and Depeweg et al. (2018), we use two toy regression problems to demonstrate the capability of CDNs of quantifying predictive uncertainty. The datasets of those probems are generated as follows.

- **Cubic** (Hernández-Lobato & Adams, 2015): We sample 20 input points $x \sim \mathcal{U}[-4, 4]$ and their target values $y = x^3 + \epsilon$, where $\epsilon \sim \mathcal{N}(0, 3^2)$.
- **Mixture** (Depeweg et al., 2018): We sample 750 input points $x \sim p(x)$, where $p(x) = \frac{1}{3}\mathcal{N}(-4, \frac{2}{5}) + \frac{1}{3}\mathcal{N}(0, 0.9) + \frac{1}{3}\mathcal{N}(4, \frac{2}{5})$ and generate their target values by $y = \sin x + 3|\cos \frac{x}{2}|\epsilon$, where $\epsilon \sim \mathcal{N}(0, 1)$.

We use a single layer MLP with 100 hidden units as the predictive network, while the hypernetworks ($g_1$ and $g_2$) are modeled with single layer MLPs with 10 hidden units each. Three samples from the mixing distribution are used to do Monte Carlo integration to approximate eq. (4).

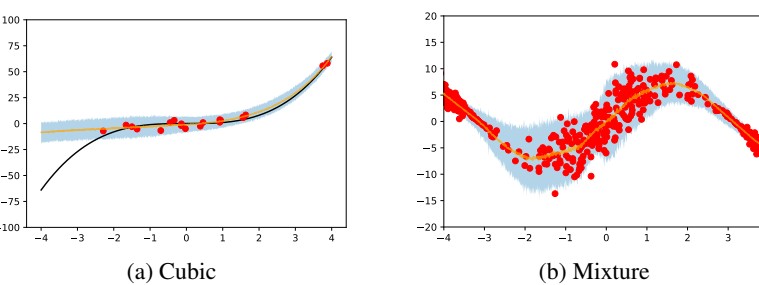

|                | (a) Cubic | (b) Mixture |
|----------------|-----------|-------------|

Figure 2: Predictive distributions of CDN. Black lines corresponds to the true noiseless function, red dots correspond to samples, orange lines and shaded regions correspond to the empirical mean and the $\pm 3$ standard deviation of the predictive distribution, respectively.

The results are presented in Figure 2. We observe that our model is able to accurately quantify the heteroscedastic noise in the Mixture dataset. Comparing the results for the CDN with those of Bayesian models on the Cubic dataset (Figure 8 in the appendix) it becomes clear that the CDN is quantifying the noise applied in the data-generating process (aleatoric uncertainty) instead of epistemic uncertainty. A visualization of the learned mixing distributions for different input points is provided in Appendix G.

## 6.2 OUT-OF-DISTRIBUTION CLASSIFICATION

Following Lakshminarayanan et al. (2017), we train all models on the MNIST training set and investigate their performance on the MNIST test set and the notMNIST dataset[12], which contains

---

[10]https://github.com/AMLab-Amsterdam/MNF_VBNN
[11]https://github.com/gd-zhang/noisy-K-FAC
[12]Available at http://yaroslavvb.blogspot.com/2011/09/notmnist-dataset.html.

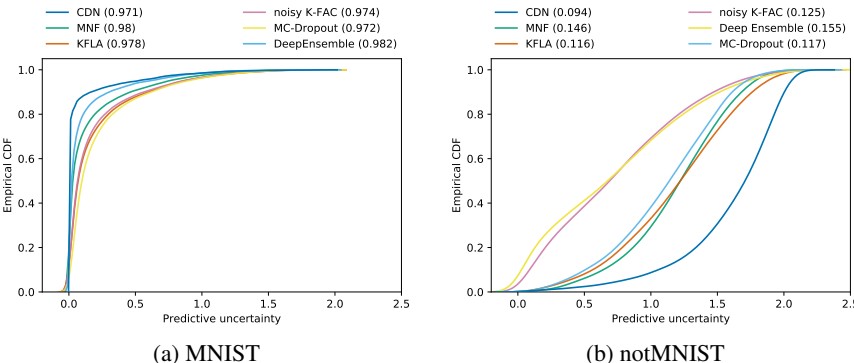

Figure 3: CDFs of the empirical entropy of the predictive distribution of the models, where the y-axis denotes the fraction of predictions having entropy less than the corresponding value on the x-axis. Confident models should have its CDF close to the top-left corner of the figure, while uncertain models to the bottom-right. The number next to each model name indicates its test accuracy.

images (of the same size and format as MNIST) of letters from the alphabet instead of handwritten digits. On such an out-of-distribution test set, the predictive distribution of an ideal model should have maximum entropy, i.e. it should have a value of $\ln 10 \approx 2.303$ which would be achieved if all ten classes are equally probable. The NN used for this experiment is an MLP with a 784-100-10 architecture.

We present the results in Figure 3, where we plotted the cumulative distribution function (CDF) of the empirical entropy of the predictive distribution, following Louizos & Welling (2017). A CDF curve close to the top-left corner of the figure implies that the model yields mostly low entropy predictions, indicating that the model is very confident. While one wishes to observe high confidence on data points similar to those seen during training, the model should express uncertainty when exposed to out-of-distribution data. That is, we prefer a model to have a CDF curve closer to the bottom-right corner on notMNIST, as this implies it makes mostly uncertain (high entropy) predictions, and a curve closer to the upper-left corner for MNIST. As the results show, our model has very high confidence on the test set of MNIST while having the lowest confidence on notMNIST compared to all baseline models. It is surprising that even though CDNs are designed to capture aleatoric uncertainty they outperforms other models in this experiment, which is rather designed for quantifying model uncertainty (Kendall & Gal, 2017).

### 6.3 ADVERSARIAL EXAMPLES

To investigate the robustness and detection performance of the CDN w.r.t. adversarial examples (Szegedy et al., 2014), we apply the Fast Gradient Sign Method (FGSM) (Goodfellow et al., 2015) to a 10% fraction (i.e. 1000 samples) of the MNIST and CIFAR-10 test set. We do so, by making use of the implementation provided by Cleverhans (Papernot et al., 2018). We use the same MLP (with 784-100-10 architecture) as before for experiments on MNIST, and the LeNet5 convolutional network (LeCun et al., 1998) for experiments on CIFAR-10. Out of implementation reasons, we only model the parameters of the fully-connected layers of LeNet5 as random variables for CDN and KFLA. The probabilistic hypernetworks are two-layer MLPs with 100 hidden units. Note, that we do not use adversarial training when training the Deep Ensemble in this experiment to allow for a fair comparison.

**MNIST** Figure 4 presents the accuracy and the average empirical entropy of the predictive distribution w.r.t. adversarial examples for MNIST with varying levels of perturbation strength (between 0 and 1). We observe that the CDN is significantly more robust to adversarial examples than all baseline models. The prediction entropy for adversarial examples is also higher for the CDN than for most other models (only MC-dropout shows higher uncertainty in the beginning). Furthermore, the prediction uncertainty of the CDN is steadily increases with increasing perturbation strength.

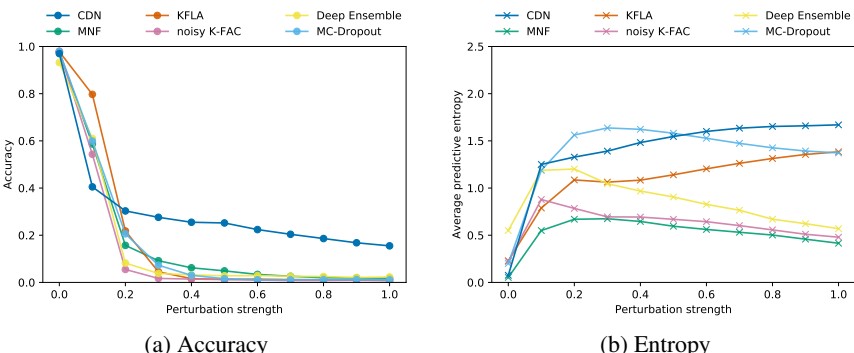

(a) Accuracy

(b) Entropy

Figure 4: Prediction accuracy and average entropy of models trained on MNIST when attacked by FGSM-based adversarial examples (Goodfellow et al., 2015) with varying perturbation strength.

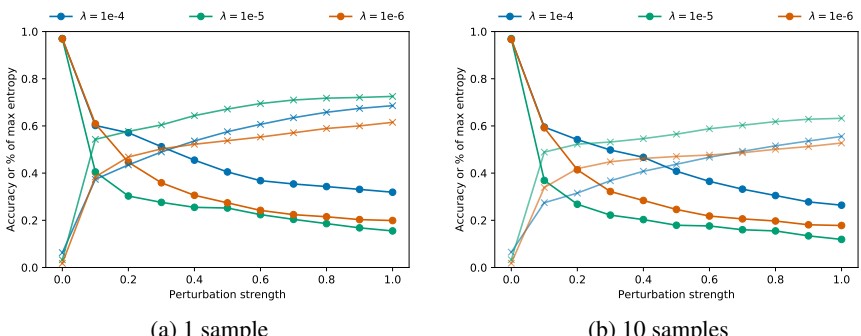

(a) 1 sample

(b) 10 samples

Figure 5: Accuracy and average entropy of our proposed model under FGSM attack. We use a single sample of adversarial example in the left figure, and use the average of 10 adversarial examples to take into account the stochasticity of our model in the right figure. Circle indicates accuracy, while cross indicates entropy. The y-axis represents both the accuracy and the relative entropy to the maximum entropy (i.e. $\ln 10$).

We demonstrate how the performance of our model depends on the choice of regularization strength $\lambda$ in Figure 5. It is clearly visible that $\lambda$ acts as a hyperparameter that balances the trade-off between detection (uncertainty) and robustness (accuracy). When allowing for lower uncertainty, the CDN gets surprisingly robust, showing a prediction accuracy of around 0.4 even for a perturbation strength of 1. Even when we increase the strength of the adversarial examples by computing them based on 10 different samples from our model, the CDN is still significantly more robust than all baseline models w.r.t. the weaker adversarial examples relying on a single sample.

**CIFAR-10** Figure 6 shows that, leading to second best results w.r.t. accuracy as well as uncertainty, our model is competitive to other state of the art models on CIFAR-10. Note, that for the CDN we do not use probabilistic hypernetworks for the convolution layers of LeNet5, i.e. we only treat the parameters of the fully-connected layers as random variables. Treating all parameters as random variables, as it is done by the BNNs in this experiment (except from KFLA) could potentially improve the performance of the CDN.

### 6.4 COMPARISON TO TRAINING BASED ON ELBO AND VIB OBJECTIVE

In this section, we experimentally investigate the effects of (a) optimizing our proposed objective (eq. (5)), (b) treating our model as a BNN and train it with the ELBO objective (eq. (10)), and (c) following the VIB approach (which corresponds to optimizing the objective which results from introducing a hyperparameter $\lambda$ in front of the KL-term in eq. (10)). As stated before, (a) and (c)

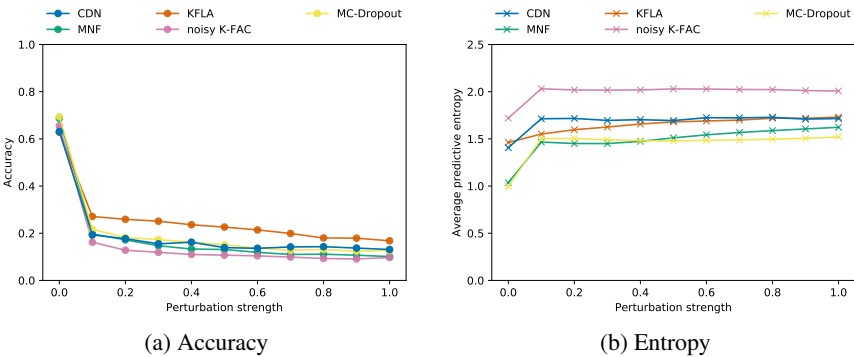

Figure 6: Prediction accuracy and average entropy of models trained on CIFAR-10 when attacked by FGSM-based adversarial examples with varying perturbation strength.

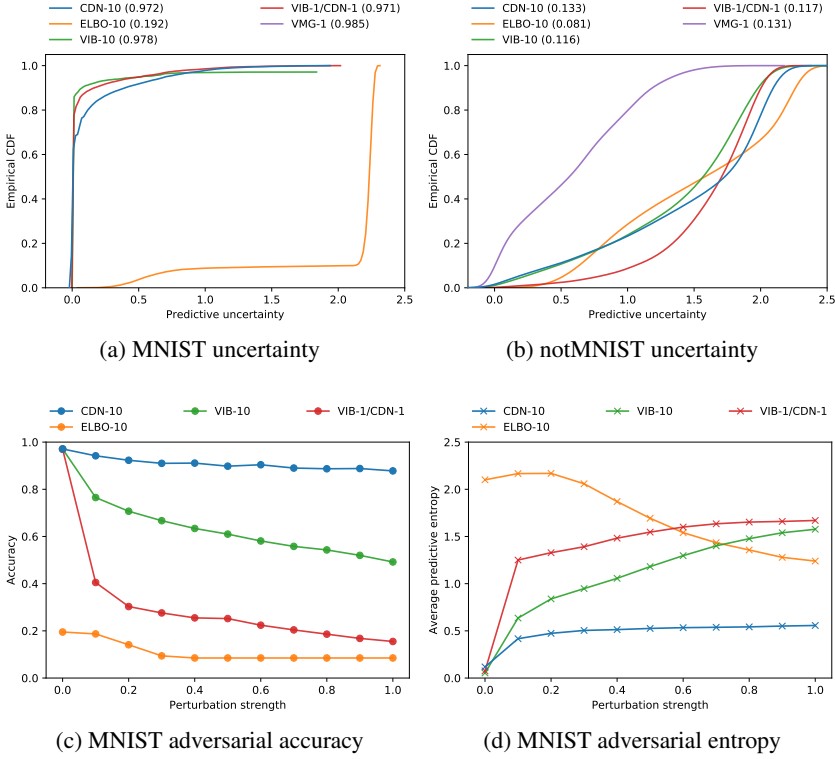

Figure 7: Comparison between the ELBO, VIB and the CDN objective. "Objective-S" denotes that the objective was approximated based on $S$ samples of $\boldsymbol{\theta}$ during training.

become equivalent when approximating the objective by a single sample of $\boldsymbol{\theta}$, and both become equivalent to (b) when setting $\lambda = 1$ in addition. To better analyze the effects of following the different objectives we therefore approximate them based on 10 samples.

Results for the OOD classification and the robustness to adversarial examples are shown in Figure 7. Training based on the ELBO results in unsatisfactory performance as can be seen from the orange graphs and the indicated low validation accuracy. The CDN and the VIB objective both work well on the OOD classification task, where no performance is gained by using 10 instead of one sample (which makes both equivalent). This suggests that the input dependency of the distribution over $\boldsymbol{\theta}$ (in addition to the weighting of the KL-term in the objective) plays a crucial role for the increased

performance observed compared to baseline models for OOD classification. This hypothesis is also supported by the fact that the performance is increase compared to the VMG (Louizos & Welling, 2016), which is a closely related BNN using a MVN distributions as approximate (input independent!) posterior.

While for the robustness against adversarial attacks the increased sample size improved the performance of models trained with the VIB as well as with the CDN objective, the model trained with the CDN objective clearly outperforms the others, reaching an surprisingly high accuracy about 0.9 even under huge perturbations.

## 7 CONCLUSION

We introduce compound density networks (CDNs), a new framework that allows for better uncertainty quantification in neural network (NN), and corresponds to a compound distribution (i.e. mixture with uncountable components) in which both the component distribution and the mixing distribution are parametrized by NNs. CDNs are inspired by the success of recently proposed ensemble methods and represent a continuous generalization of mixture density networks (MDNs) (Bishop, 1994). They can be implemented by using a hypernetwork to map the input to a distribution over the parameters of the target NN, that models a predictive distribution. An extensive experimental analysis showed that CDNs are able to produce promising results in terms of uncertainty quantification. Especially, when facing FGSM-based adversarial attacks, the predictions of our model are significantly more robust than those of previous models, while simultaneously providing a better chance of detecting the attack by showing increased uncertainty.

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

## APPENDIX A   PSEUDOCODE FOR TRAINING CDNS

---

**Algorithm 1** The training procedure of CDN.

---

**Require:**
   Mini-batch size $M$, number of samples $S$ used for Monte Carlo integration (eq. (4)), regularization strength $\lambda$, and learning rate $\alpha$.

1: **while** the stopping criterion is not satisfied **do**
2:     $\{\mathbf{x}_m, \mathbf{y}_m\}_{m=1}^M \sim \mathcal{D}$                                    ▷ Sample mini-batch from dataset
3:     **for** $m = 1, \ldots, M; s = 1, \ldots, S$ **do**
4:         $\boldsymbol{\theta}_{ms} \sim p(\boldsymbol{\theta}|g(\mathbf{x}_m; \boldsymbol{\psi}))$                          ▷ Use reparametrization trick
5:         $\boldsymbol{\phi}_s(\mathbf{x}_m) = f(\mathbf{x}_m; \boldsymbol{\theta}_{ms})$
6:     **end for**
7:     $\log p(\mathcal{D}|\boldsymbol{\psi}) = \sum_{m=1}^M \log \frac{1}{S} \sum_{s=1}^S p(\mathbf{y}_m|\boldsymbol{\phi}_s(\mathbf{x}_m))$     ▷ Eq. 4 with Monte Carlo integration
8:     $D_{\text{KL}} = \sum_{m=1}^M D_{\text{KL}}[p(\boldsymbol{\theta}|g(\mathbf{x}_m; \boldsymbol{\psi}))\|p(\boldsymbol{\theta}))]$
9:     $\mathcal{L}(\boldsymbol{\psi}) = \log p(\mathcal{D}|\boldsymbol{\psi}) - \lambda D_{\text{KL}}$
10:     $\boldsymbol{\psi} \leftarrow \boldsymbol{\psi} + \alpha \nabla \mathcal{L}(\boldsymbol{\psi})$                                    ▷ Update parameters $\boldsymbol{\psi}$
11: **end while**

---

## APPENDIX B   CDNS WITH LATENT VARIABLES

In Section 3, we introduce CDNs as a compound distribution where the stochasticity of $f(\mathbf{x}; \boldsymbol{\theta})$ comes from treating $\boldsymbol{\theta}$ as random variables. Another approach to achieve stochasticity on $f$ is by introducing latent variable $\mathbf{z}$ and letting the parameter $\boldsymbol{\theta}$ to be fixed (learned with MLE). The CDN would then be descibed by

$$p(\mathbf{y}|\mathbf{x}) = \int p(\mathbf{y}|\mathbf{x}, \mathbf{z}; \boldsymbol{\theta})p(\mathbf{z}|\mathbf{x})d\mathbf{z}$$

$$:= \int p(\mathbf{y}|f(\mathbf{x}; \mathbf{z}, \boldsymbol{\theta}))p(\mathbf{z}|g(\mathbf{x}; \boldsymbol{\psi}))d\mathbf{z}$$

$$= \mathbb{E}_{p(\mathbf{z}|g(\mathbf{x};\boldsymbol{\psi}))}[p(\mathbf{y}|f(\mathbf{x}; \mathbf{z}, \boldsymbol{\theta}))] \ . \tag{11}$$

The log-likelihood function is therefore

$$\log p(\mathcal{D}|\boldsymbol{\theta}, \boldsymbol{\psi}) = \sum_{n=1}^N \log \mathbb{E}_{p(\mathbf{z}|g(\mathbf{x}_n;\boldsymbol{\psi}))}[p(\mathbf{y}_n|f(\mathbf{x}_n; \mathbf{z}, \boldsymbol{\theta}))] \ , \tag{12}$$

and the objective function is

$$\mathcal{L}(\boldsymbol{\theta}, \boldsymbol{\psi}) = \log p(\mathcal{D}|\boldsymbol{\theta}, \boldsymbol{\psi}) - \lambda D_{\text{KL}}[p(\mathbf{z}|g(\mathbf{x}_n; \boldsymbol{\psi}))\|p(\mathbf{z})] \ . \tag{13}$$

The training algorithm in Algorithm 1 only needs to be trivially modified to accommodate $\boldsymbol{\theta}$, i.e. in the gradient steps. In the following section, we present an instance of this framework.

### B.1   A VARIANT OF ADAPTIVE DROPOUT

This model relies on injecting noise into the hidden activations of the neural network $f$, similar to dropout (Srivastava et al., 2014). However, in this model, we condition the dropout's Gaussian on the activations of the previous hidden layer, similar to adaptive Bernoulli dropout proposed by Ba & Frey (2013).

Let $\mathbf{z}_l \sim p(\mathbf{z}_l|g_l(\mathbf{h}_{l-1}; \boldsymbol{\psi}_l))$ be the noise vector of layer $l$. We then multiplicatively apply this noise to the activations $\mathbf{h}_l$, i.e. $\mathbf{h}_l \odot \mathbf{z}_l$, where $\odot$ denotes the Hadamard product. Let $\mathbf{z} := \{\mathbf{z}_l\}_{l=1}^{L-1}$ and let $g(\mathbf{x}; \boldsymbol{\psi}) := \{g_l(\mathbf{h}_{l-1}; \boldsymbol{\psi}_l)\}_{l=1}^L$. Let $p(\mathbf{z}|g(\mathbf{x}; \boldsymbol{\psi})) =: \prod_{l=1}^{L-1} p(\mathbf{z}_l|g(\mathbf{h}_l; \boldsymbol{\psi}_l))$ be the mixing distribution. Each of the $p(\mathbf{z}_l|g(\mathbf{h}_l; \boldsymbol{\psi}_l))$ can be any distribution as long as we can apply reparametrization trick. Then, we can immediately apply latent CDNs training procedure on this model. One example of $p(\mathbf{z}_l|g_l(\mathbf{h}_{l-1}; \boldsymbol{\psi}_l))$ is $\mathcal{N}(\mathbf{z}_l|1, g_l(\mathbf{h}_{l-1}; \boldsymbol{\psi}_l))$, which then the model resembles Gaussian dropout (Srivastava et al., 2014) with adaptive scaling.

## APPENDIX C    DETAILS ABOUT THE MATRIX VARIATE NORMAL (MVN)

Let $\mathbf{X} \in \mathbb{R}^{r \times c}$ and $p(\mathbf{X}) := \mathcal{MN}(\mathbf{X}|\mathbf{M}, \text{diag}(\mathbf{a}), \text{diag}(\mathbf{b}))$. Taken from Louizos & Welling (2016), the procedure to sample from $p(\mathbf{X})$ using reparametrization trick (Kingma & Welling, 2014) and the closed-form expression of the KL-divergence to $\mathcal{MN}(\mathbf{0}, \mathbf{I}, \mathbf{I})$ are presented in the following sections.

### C.1    REPARAMETRIZATION TRICK

Let $\mathcal{E} \in \mathbb{R}^{r \times c}$. Sampling $p(\mathbf{X})$ can be done by

$$\mathcal{E} \sim \mathcal{MN}(\mathbf{0}, \mathbf{I}, \mathbf{I}) \iff \epsilon_{ij} \sim \mathcal{N}(0, 1) \ \forall\, i = 1, \ldots, r \, \forall\, j = 1, \ldots, c \tag{14}$$

$$\mathbf{X} = \mathbf{M} + \text{diag}(\mathbf{a})^{\frac{1}{2}} \mathcal{E} \, \text{diag}(\mathbf{b})^{\frac{1}{2}} \tag{15}$$

### C.2    KL-DIVERGENCE TO STANDARD MVN

Let $\mathbf{I}_r \in \mathbb{R}^{r \times r}$, $\mathbf{I}_c \in \mathbb{R}^{c \times c}$ be identity matrices, the KL-divergence between $p(\mathbf{X})$ and $\mathcal{MN}(\mathbf{X}|\mathbf{0}, \mathbf{I}, \mathbf{I})$ is given by

$$D_{\text{KL}}[p(\mathbf{X}) \| \mathcal{MN}(\mathbf{0}, \mathbf{I}, \mathbf{I})] = \frac{1}{2} \left( \sum_{i=r} a_i \sum_{j=1}^{c} b_j + \|\mathbf{M}\|_F^2 - rc - c \sum_{i=1}^{r} \log a_i - r \sum_{j=1}^{c} \log b_j \right) . \tag{16}$$

## APPENDIX D    VECTOR SCALING PARAMETRIZATION

The naive formulation of $g_l$ can be very expensive in term of number of parameters. Suppose $\mathbf{W}_l \in \mathbb{R}^{r \times c}$ and $g_l$ is a two layer MLP with $k$ hidden units. Then $g_l$ would have $rk + krc + kr + kc$ many parameters, which quickly becomes very large for a moderately sized NNs. The majority of the parameters are needed to define the mean matrix $\mathbf{M}_l$. Following the approach of Ha et al. (2017) and Krueger et al. (2017), we make a trade-off between expressiveness of $g_l$ on the mean matrix with the number of parameter by instead replacing $\mathbf{M}_l$ with a matrix $\mathbf{V}_l$ of the same size and a vector $\mathbf{d}_l \in \mathbb{R}^r$, which is the output of $g_l$. Thus, now $g_l$ maps $\mathbf{h}_{l-1} \longmapsto \{\mathbf{d}_l, \mathbf{a}_l, \mathbf{b}_l\}$ and we can get $\mathbf{M}_l$ by

$$\mathbf{M}_l^f = \begin{bmatrix} d_{l1}^f \mathbf{v}_{l1} \\ d_{l2}^f \mathbf{v}_{l2} \\ \ldots \\ d_{lr}^f \mathbf{v}_{lr} \end{bmatrix} . \tag{17}$$

That is, each element of $\mathbf{d}_l$ is being used to scale the corresponding row of $\mathbf{V}_l$. Note that although $\mathbf{V}_l$ is a parameter matrix with the same size of $\mathbf{M}_l$, it crucially is not an output of $g_l$ as in the naive parametrization. Thus the number of parameter of $g_l$ is now $rk + rc + 2kr + kc$, which is more manageable and implementable for larger weight matrices.

## APPENDIX E    EXPERIMENTAL RESULTS FOR ADDITIONAL MODELS

In this section we compare the CDN descibed in the main text with the CDN proposed in Appendix B.1 (CDN-dropout), the BNN proposed by Louizos & Welling (2016) (VMG), a mixture of experts (MoE) and an MDN on the MNIST dataset. For both MDN and MoE, we use two-layer MLP with 100 hidden units, with 5 mixture components. Specifically for MoE, the mixing distribution is also given by another NN of the same architecture. We use the code provided by Louizos & Welling (2016) to get VMG's results. [13] The results for the out-of-distribution prediction are presented in Figure 9. Note that the analysis presented in the main text applies here as well.

---

[13]https://github.com/AMLab-Amsterdam/SEVDL_MGP.

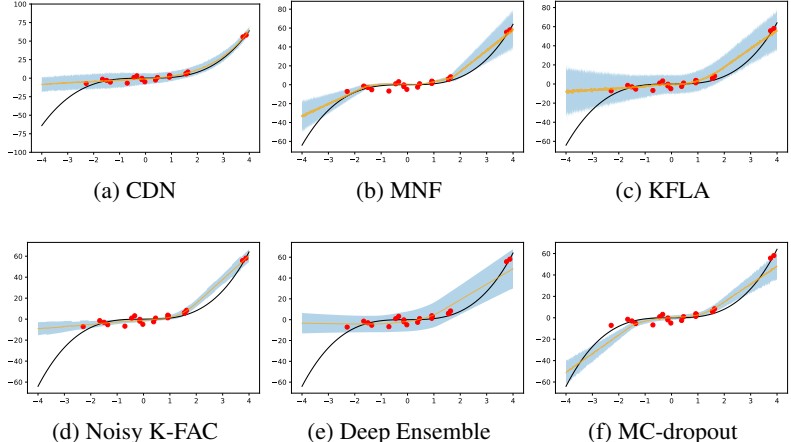

(a) CDN       (b) MNF       (c) KFLA

(d) Noisy K-FAC   (e) Deep Ensemble   (f) MC-dropout

Figure 8: Comparison of the predictive distributions given by the CDN and the baselines on toy dataset introduced by Hernández-Lobato & Adams (2015). Black lines corresponds to the true noiseless function, red dots correspond to samples, orange lines and shaded regions correspond to the empirical mean and the $\pm 3$ standard deviation of the predictive distribution, respectively.

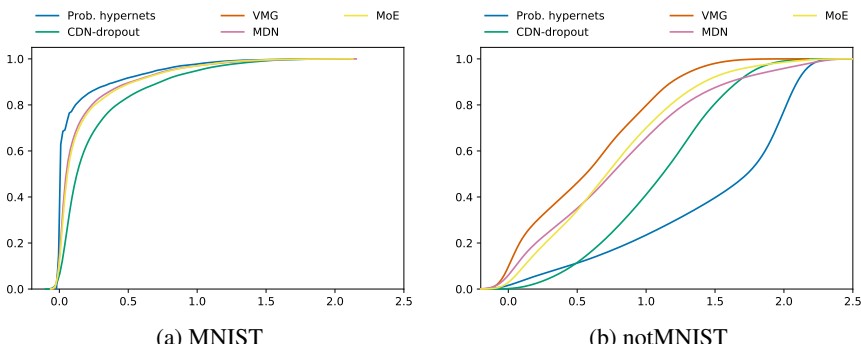

(a) MNIST             (b) notMNIST

Figure 9: CDF of the prediction entropy on MNIST and notMNIST test set of additional models.

## APPENDIX F    HYPERPARAMETERS

For the baseline models in our experiments, we use the hyperparameter values that are suggested in the respective publications ans summarized in the following:

- **MNF:** The KL-term is weighted by $1/B$, where $B$ is the number of mini-batches used during optimization (see (Graves, 2011) for a justification of this). Moreover it is annealed with a hyperparameter initialized to 0 and increasing to 1 during training. We found that this yields better results than using no re-weighting of the KL term and is necessary to achieve the results reported by Louizos & Welling (2017).

- **noisy K-FAC:** KL-term weight is set to $\lambda = 0.5$ and the prior variance to $\eta = 0.01$.

- **KFLA:** we use the best prior precision $\tau \in \{1, 10, 20, 30, 40, 50\}$ we found in an hyperparameter search, which we detail in the next section.

- **Deep Ensemble:** The number of mixture components is 5, the adversarial perturbation strength is set to 1% of the input range, and the weight decay is 0.001.

- **MC-dropout:** The dropout probability was set to 0.5 and the weight decay parameter to 0.001.

## F.1 HYPERPARAMETER SEARCH

We perform a hyperparameter search on $\lambda$ for the training of CDNs and for $\tau$ for KFLA based on the MNIST validation accuracy, and pick the highest $\lambda$ (the lowest $\tau$) that achieve $> 0.97$ validation accuracy.

| $\lambda$ | CDN |
|---|---|
| 1 | 0.107 |
| 0.1 | 0.189 |
| 0.01 | 0.111 |
| 0.001 | 0.948 |
| 0.0001 | 0.966 |
| 0.00001 | 0.972 |
| 0.000001 | 0.972 |

| $\tau$ | KFLA |
|---|---|
| 1 | 0.806 |
| 10 | 0.977 |
| 20 | 0.978 |
| 30 | 0.978 |
| 40 | 0.978 |

Table 1: Validation accuracy on MNIST vs $\lambda$ for CDN and $\tau$ for KFLA.

| $\lambda$ | MNF | n. K-FAC | Deep Ens. | MC-dropout |
|---|---|---|---|---|
| 1 | 0.981 | 0.974 | 0.207 | 0.162 |
| 0.1 | 0.981 | 0.976 | 0.885 | 0.834 |
| 0.01 | 0.978 | 0.981 | 0.965 | 0.932 |
| 0.001 | 0.98 | 0.981 | 0.983 | 0.971 |
| 0.0001 | 0.98 | 0.979 | 0.986 | 0.976 |
| 0.00001 | 0.98 | 0.979 | 0.986 | 0.977 |

Table 2: Validation accuracy on MNIST vs $\lambda$.

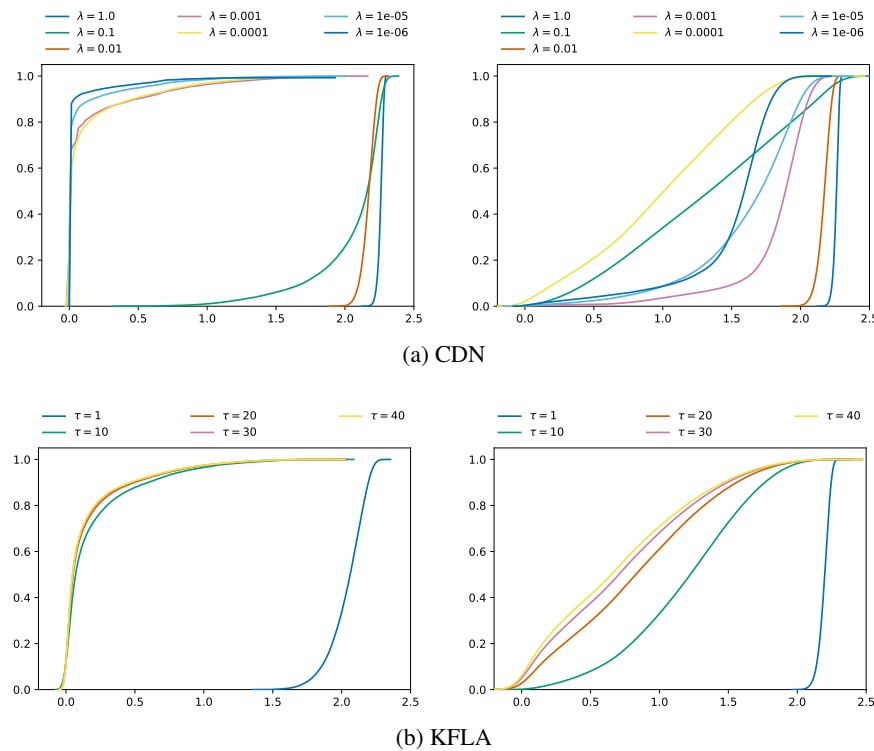

(a) CDN

(b) KFLA

Figure 10: Empirical CDF curve of uncertainty on MNIST (left) and notMNIST (right)

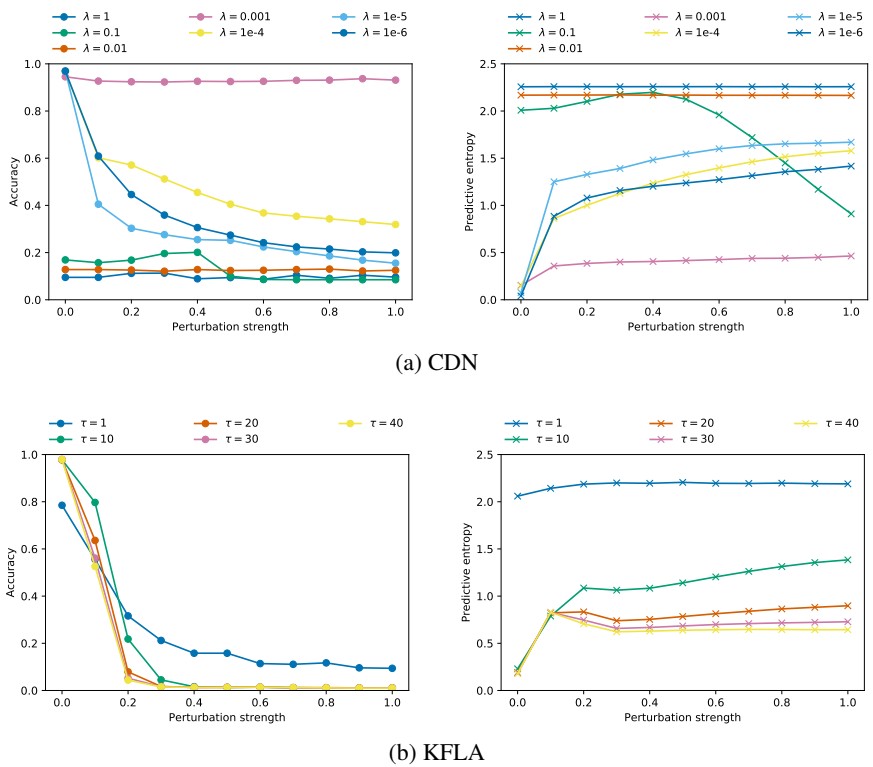

(a) CDN

(b) KFLA

Figure 11: Accuracy (left) and uncertainty (right) w.r.t. adversarial examples.

## APPENDIX G    VISUALIZATION OF THE LEARNED MIXING DISTRIBUTION

To further understand the effect of conditioning the distribution over $\boldsymbol{\theta}$ (i.e. the mixing distribution) we compute the mixing distribution $p(\boldsymbol{\theta}|g(\mathbf{x}_i; \boldsymbol{\psi}))$ for a set of samples $\mathbf{x}_1, \ldots \mathbf{x}_n$ and a CDN trained on a variant of the Cubic dataset, where we add random noise of different variance depending on $\mathbf{x}$ (that is, $\epsilon \sim \mathcal{N}(0, 15^2)$ if $\mathbf{x} < 0$ and $\epsilon \sim \mathcal{N}(0, 3^2)$, otherwise). The results for two randomly selected weights $w_i^{(l)} \in \boldsymbol{\theta}$ are shown in Figure 12. We found that the mean and the variance of the distribution (which is Gaussian due to our model) varies depending on the value of the input $\mathbf{x}$ indicating that different mixture components get high probability for different inputs.

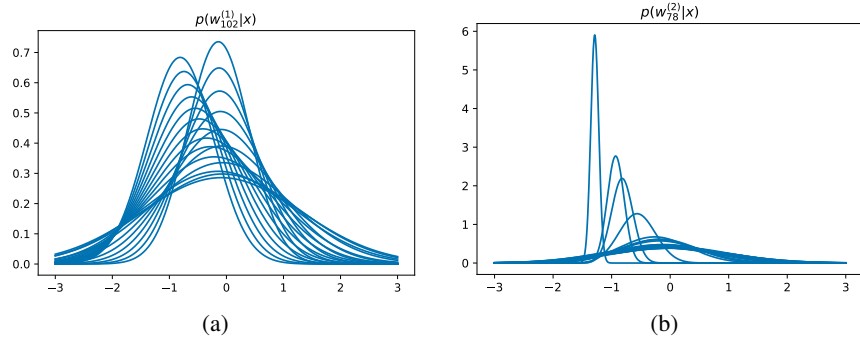

(a)                     (b)

Figure 12: Visualization of the distribution $p(w_i^{(l)}|g(\mathbf{x}; \boldsymbol{\psi}))$ of a randomly selected weight $w_i^{(l)} \in \boldsymbol{\theta}$ for different samples of input $\mathbf{x}$ from the modified Cubic toy dataset. $w_i^{(l)}$ denotes the $i$-th weight of the $l$-th layer of $f(\mathbf{x}; \boldsymbol{\theta})$.

Furthermore, to show that CDNs are able to capture multimodality in weight space, we train a CDN with a 5 hidden units mixture component on a toy classification dataset that is constructed as follows: we sample an input $\mathbf{x}$ from a mixture of Gaussian $p(\mathbf{x}) = \frac{1}{2}\mathcal{N}(-3, 1) + \frac{1}{2}\mathcal{N}(3, 1)$, and assign a label depending whether it comes from the first ($\mathbf{y} = 0$) or the second Gaussian ($\mathbf{y} = 1$). To evaluate the resulting distribution, we marginalize the mixing distribution $p(\boldsymbol{\theta}|g(\mathbf{x}; \boldsymbol{\psi}))$ w.r.t. $\mathbf{x}$, i.e. we evaluate $p(\boldsymbol{\theta}) = \int p(\boldsymbol{\theta}|g(\mathbf{x}; \boldsymbol{\psi}))p(\mathbf{x}) \, d\mathbf{x}$. The resulting distribution for two randomly selected weights $w_i^{(l)} \in \boldsymbol{\theta}$ are shown in the figure below. We observe that indeed our model can learn a multimodal weight distribution.

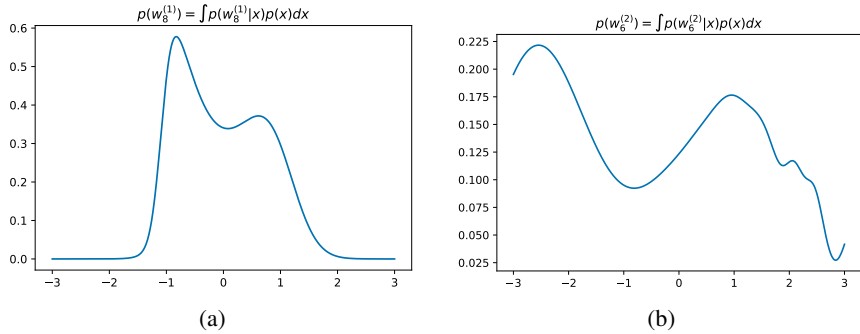

(a)                         (b)

Figure 13: Visualization of the marginal distribution $p(w_i^{(l)}) = \int p(w_i^{(l)}|g(\mathbf{x}; \boldsymbol{\psi}))p(\mathbf{x}) \, d\mathbf{x}$ for two randomly selected weights $w_i^{(l)}$ of a CDN trained on a toy classification dataset.

