# OpenReview forum: "Compound Density Networks"
_ICLR.cc/2019/Conference_

### Official Review · AnonReviewer2 · 2018-11-02
**potentially interesting ideas; could be better justified and validated.**

**Rating:** 5
**Confidence:** 4

**Review:**

The paper addresses the problem of producing sensible (high) uncertainties on out of distribution (OOD) data along with accurate predictions on in-distribution data. The authors consider a model wherein the weights of the network (\theta) are drawn from a matrix normal distribution whose parameters are in-turn a (non-linear; parameterized by a another network) function of the covariates (x). Instead of inferring a posterior over theta that then induces the predictive uncertainties, uncertainties here arise from a regularizer that penalizes the distribution over theta from deviating too far from a standard Normal. Experiments present results on toy data, MNIST/not MNIST as well as on adversarial perturbations of MNIST and CIFAR 10 datasets.

The paper is clearly written and addresses an important problem. The paper presents both an alternate model as well as an alternate objective function. While the authors do report some interesting results, they do a poor job of motivating the proposed extensions. It isn’t clear why the particular proposals are necessary or to which of the proposed extensions the inflated OOD uncertainties can be attributed:

1. The proposed model? Is using a conditional weight prior p(\theta | x) (Eq 3) instead of p(\theta) (as in BNNs)  necessary for the inflated uncertainties on OOD data?

2. The  objective? The proposed objective,  Eq 5, trades off stochastically approximating the (conditional) marginal likelihood against not deviating too much from p(\theta) =  MN(0, I, I) in the KL sense. Depending on \lambda, the objective either closely approximates the marginal likelihood or not. It is unclear how important this particular objective is to the results.
    -  Instead of relying on the KL regularizer, a standard approach to learning the model in Eq 3 would be to use well understood MCMC or variational methods that explicitly retain uncertainty in \theta and induce predictive uncertainties.  Were they explored and found to be not effective? It would be nice to see how a “gold standard” HMC based inference does on at least the small toy problem of Sec 5.1?
    - There is also a closely related variant of Eq 3 which we can arrive at by switching the log and the expectation in the first term of Eq 5 and applying Jensen’s inequality —> E_p(\theta| x)[ln p(y | x, \theta)] - KL (p(\theta | x) || p(\theta)). This would correspond to maximizing a valid lower bound to the marginal likelihood of a BNN model p(y | x, \theta) p(\theta), while interpreting p(\theta | x) as an amortized variational approximation. This variant has the advantage that it provides a valid lower bound on the marginal likelihood, and exploits the well understood variational inference machinery. This also immediately suggests, that the variational approximation , p (\theta | x)  should probably depend on both x and y rather than only on x and the flexibility of the hyper networks g would govern how well the true posterior over weights \theta can be approximated.
Comparisons against these more standard inference algorithms is essential for understanding what advantages are afforded by the objective proposed in the paper.

3.  Or simply to a well tuned \lambda, chosen on a per dataset basis? From the text it appears that \lambda is manually selected to trade off accuracy against uncertainty on OOD data. In the real world, one would not have access to OOD data during training, how is one to pick \lambda in such cases?

Detailed comments about experiments:

a) The uncertainties produced by CDN in Figure 2 seems strange. Why does it go to nearly zero around x = 0, while being higher in surrounding regions with more data?

b) Down weighting the KL term by lambda for the VI techniques unfairly biases the comparison. This forces the VI solution to tend to the MLE, sacrificing uncertainty in the variational distribution. It would be good to include comparisons against VI with \lambda = 1.

==========

There are potentially interesting ideas in this paper. However, as presented these ideas are poorly justified and careful comparisons against sensible baselines are missing.

---

> ### Author Response · Authors · 2018-11-23
> **Response to AnonReviewer2**
>
> We thank the reviewer for the valuable feedback! The suggestion comments were very helpful and led to a clear improvement of our manuscript.  We reply to the answers and comments in the order they were raised:
>
> (1) If one uses the same matrix-variate normal distribution that we use for p(\theta | x) as approximate posterior p(\theta) of a BNN in conjunction with the ELBO objective, one arrives at a BNN proposed by Louizos and Welling (2016) [1], i.e. the Variational Matrix Gaussian (VMG). We found that VMG’s results (obtained from their original code https://github.com/AMLab-Amsterdam/SEVDL_MGP) are not as good as that for the CDN, as shown in Figure 8 in the appendix. This is further discussed in the newly added section 6.4.
>
> (2) Thank you for this valuable suggestion! We have added a new section (Sec. 4) to discuss the differences between the objective used for CDN, when performing variational inference for BNNs, and in the variational information bottleneck (VIB) framework. Furthermore, we present an experimental investigation of these different objectives (Sec. 6.4). We found that the CDN objective leads to superior results, especially in the adversarial examples experiment.
>
> (3) We observed that as \lambda increases, in the validation set, the uncertainty is increasing, while the accuracy is decreasing. So, a simple heuristic that we use is to choose the highest \lambda that allow high validation accuracy (e.g. > 0.97 on MNIST).  We found that this heuristic works very well in our experiments (the results have updated to reflect on this heuristic). We have made this procedure clear in the revised manuscript.
>
> Detailed comments about experiments:
> (a) Thanks for catching this. Indeed this was due to a bug in the toy regression experiment which we have fixed now.
> (b) We have revised the baselines so that they either use \lambda = 1 or the settings that the original authors recommended. We detail this in Appendix F.
>
>
> References:
> [1] Louizos, Christos, and Max Welling. "Structured and efficient variational deep learning with matrix gaussian posteriors." International Conference on Machine Learning. 2016.

---

> > ### Comment · AnonReviewer2 · 2018-11-27
> > **Concerns remain**
> >
> > Thank you for the rebuttal and updating the paper. Admittedly it is in much better shape. The new experiments, however, do suggest that performance crucially depends on \lambda (based on the poor performance of VI, but significantly improved performance of VIB) and I am not convinced that the proposed ad-hoc procedure for selecting \lambda generalizes, especially to datasets not as well studied as MNIST/CIFAR.

---

> > > ### Author Response · Authors · 2018-11-30
> > > **Response to AnonReviewer2**
> > >
> > > Thank you very much for the fast response! Indeed the results depend on the right choice of \lambda. Note, however that this is also the case for the VIB model proposed by Alemi, et al. (2017) (see e.g. fig. 5 and 6), which is very related to our model as we pointed out in the revised manuscript. We do not see any reason, why this hyperparameter selection problem should be any harder for a CDN than it is for a VIB and why the validation set based selection poses a problem that speaks against acceptance.
> > >
> > > Moreover, we have run another experiment in which we trained a CDN on FashionMNIST (https://github.com/zalandoresearch/fashion-mnist) and performed an OOD test on the flipped FashionMNIST’s test set (both up-down and left-right), MNIST, and notMNIST. We found that using the same selection heuristic for \lambda, worked fine on this  dataset as well:
> > >
> > > Hyperparameter search on FashionMNIST
> > > \lambda; val. acc; avg. entropy on: flipped up-down FashionMNIST; flipped left-right FashionMNIST; MNIST; notMNIST
> > > 1e-4; 0.848; 0.975; 1.397; 0.908
> > > 1e-5; 0.87; 0.863; 1.424; 0.741
> > > 1e-6; 0.887; 0.722; 0.978; 0.469
> > >
> > > We set accuracy threshold to be 0.87 (as a vanilla NN achieves approximately this accuracy). Based on our heuristic, we pick \lambda = 1e-5. Note, that we can clearly see that accuracy is increasing is while entropy is decreasing as \lambda decreases, which is in line with our rationale of proposing our heuristic.
> > >
> > > References:
> > > [1] Alemi, Alexander A., et al. "Deep variational information bottleneck." ICLR 2017.

---

### Official Review · AnonReviewer1 · 2018-11-02
**Clearly presented idea that needs a bit more work. Experiments show some benefit over the baselines but, overall, are not very convincing.**

**Rating:** 5
**Confidence:** 4

**Review:**

This paper proposed Compound Density Networks (CDNs), a neural network architecture that parametrises conditional distributions as infinite mixtures, thus generalising the traditional finite mixture density networks (MDNs). The authors realise CDNs by treating the weights of each neural network layer probabilistically, and letting them be matrix variate Gaussians (MVGs) with their parameters given as a function of the layer input via a hypernetwork. CDNs can then be straightforwardly optimised with SGD for a particular task by using the reparametrization trick. The authors further argue that in case that overfitting is present at CDNs, then an extra KL-divergence term can be employed such that the input dependent MVG distribution is close to a simple prior that is input agnostic. They then proceed to evaluate the predictive uncertainty that CDNs offer on three tasks: a toy regression problem, out-of-distribution example detection on MNIST/notMNIST and adversarial example detection on MNIST and CIFAR 10.

The objective of this work is to provide a method for better uncertainty estimates from deep learning models. This is an important research area and relevant for ICLR. The paper is generally well written with a clear presentation of the proposed model. The generalisation from the finite MDN to the continuous CDN seems straightforward, the model is relatively easy to implement and it is evaluated extensively against several modern baselines. Nevertheless, I believe that it still has to address some points in order to be better suited for publication:

- It seems that the model is not very scalable; while the authors do provide a way of reducing the necessary parameters that the hypernetwork has to predict, minibatching can still be an issue as it is implied that you draw a separate random weight matrix for each datapoint due to the input specific distribution (as shown at Algorithm 1). Is this how you implemented minibatching in practice? How easily is this applied to convolutional architectures?

- How many samples did you use from p(theta|x) during training? It seems that with a single sample the method becomes an instance of VIB [1], only considering the weights of the network as latent variables rather than the hidden units.

- The experiments were entirely focused on uncertainty quality but we are always interested in both performance on the task at hand as as well as good uncertainty estimates. What was the performance based on e.g. classification accuracy on each of these tasks compared to the baselines? I believe that including these results will strengthen the paper and provide a more complete picture.

- Have you checked / visualised what type of weight distributions do CDNs capture? It would be interesting to see if e.g. the marginal (across the dataset) weight distribution at each layer has any multimodality as that could hint that the network learns to properly specialise to individual data points.

- The authors mention that in order to avoid overfitting they add an extra (weighted) KL-divergence term to the log-likelihood of the dataset, that encourages the weight distributions for specific points to be close to simple priors. How influential is that extra term to the uncertainty quality that you obtain in the end? How does this term affect the learned distributions in case of CDNs? Furthermore, the way that CDNs are constructed seems to be more appropriate at capturing input specific uncertainty (i.e. aleatoric) rather than global uncertainty about the data (i.e. epistemic). I believe that for the specific uncertainty evaluation tasks this paper considers the latter is more appropriate. More discussion on both of these aspects can help in improving this paper.

- As a final point; the hyper parameters that were tuned for the MNF, noisy K-FAC and KFLA baselines are not on common ground. For noisy K-FAC and MNF the lambda (which should be fixed to 1 for a correct Bayesian model) was tuned and in general lower than 1 lambdas lead to models that are overconfident and hence underperform on uncertainty tasks. For KFLA a hyper parameter “tau” was tuned; this hyperparameter instead corresponds to the precision of the Gaussian prior on the parameters. In this case, KFLA always optimises a “correct” Bayesian model for every value of the hyperparameter whereas MNF and noisy K-FAC do not. Thus I believe that it would be better if you consider the same hyper parameter on all of these methods, e.g. the precision of the Gaussian prior.

[1] Deep Variational Information Bottleneck

---

> ### Author Response · Authors · 2018-11-23
> **Response to AnonReviewer1**
>
> We thank the reviewer for the valuable feedback! The suggestion comments were very helpful and led to a clear improvement of our manuscript.  We reply to the answers and comments in the order they were raised:
>
> (1) While indeed we need more samples of weight matrices than e.g. for applying VI for BNNs for due to the input dependency, we do not believe this makes our method unscalable to real world scenarios. Note, that input dependent samples are also needed in the variational training of VAEs (where the number of hidden variables is of course much smaller than the number of weight parameters in our setting). While we present the training algorithm naively in an online version for clearness in Algorithm 1, in practice mini-batching can be done efficiently, due to the availability of batched linear algebra operations, at least in the framework we use (PyTorch), e.g. torch.bmm, broadcasting semantics, etc. For convolution layers, we can simply use a different type of mixing distribution, e.g. a fully-factorized multivariate normal instead of matrix-variate normal.
>
> (2) Thank you very much for the pointer to VIB! We have added a section in the updated manuscript to compare the objective of CDNs with that used in VIB and VI for Bayesian neural networks (see new Section 4). Furthermore, while we  always used 1 sample during training in the original submission (which indeed makes the CDN an instance of VIB) we now added experiments using 10 samples (see Section 6.4) in an experimental analysis of the different objectives. The results show that the CDN objective produces superior results compared to VI and VIB.
>
> (3) Of course! We have moved the test accuracy (which previously was only given in the Appendix and thus hard to find) to the legends of the plots to make it more easily accessible. CDNs give better uncertainty estimates while still having similar predictive power compared to the baselines.
>
> (4) Thank you for the great suggestion. We performed the following 2 experiments for the revised version: First, we picked a weight of a CDN trained on a toy regression experiment (with heteroscedastic  noise) at random and visualized its conditional distributions given different values of x. We found that the means and variances vary for different x.  Furthermore, we picked a weight of a CDN trained on a toy classification dataset (created by sampling x ~ 1/2*N(-3, 1) + 1/2*N(3, 1), and assign y=0 if x comes from the first Gaussian and y=1, otherwise) at random and visualized its marginal distributions. We found that CDNs indeed capable of learning multimodal weight distribution and to learn input specific mixing distributions.. We detail this in Appendix G.
>
> (5) We found that the regularization term has a significant impact on the quality of the prediction and the uncertainty estimate (we found that the uncertainty estimates are worse with small \lambda). It makes sure that the variance of \theta is not shrinking too much, i.e. encouraging the mixing distribution to be close to the prior implies it should have similar variance to the prior (which was chosen to be large). Naturally, the coefficient \lambda controls this behavior: as \lambda increases the validation accuracy is decreasing while the uncertainty is increasing (and vice versa). This gives rise to the selection heuristic for \lambda we applied: pick the highest \lambda that still gives high accuracy on the validation set (e.g. > 0.97 in MNIST). We found that this works very well in the experiments we did (on OOD and adversarial examples). Furthermore, indeed CDNs are rather designed to capture the (heteroscedastic) aleatoric uncertainty. We have revised the toy experiments to better account for that. However, curiously, CDNs also work well in tasks that are usually shown as prime examples of epistemic uncertainty, e.g. OOD classification and adversarial attack.
>
> (6) Thank you for this feedback. You are right! We have revised the baseline experiments with Bayesian models so that they either use \lambda = 1 or the settings that the original authors recommended, i.e. we only tune \tau in KFLA and set \tau = 0.01 in noisy-KFAC as these are the settings suggested in their respective publications. Note, that the conclusions keep unchanged.
>
>
> References:
> [1] Louizos, Christos, and Max Welling. "Structured and efficient variational deep learning with matrix gaussian posteriors." International Conference on Machine Learning. 2016.
> [2] Kingma, Diederik P., Tim Salimans, and Max Welling. "Variational dropout and the local reparameterization trick." Advances in Neural Information Processing Systems. 2015

---

> > ### Comment · AnonReviewer1 · 2018-11-27
> > **Thank you for addressing my comments but I still believe the paper needs a bit more work before publication.**
> >
> > Thank you for clarifying and revising the submission. I share a similar feeling with the other reviewers; the submission is in a better shape but it still needs a bit more work in order to be ready for publication.
> >
> > Regarding scalability; it would be interesting to report and compare runtimes on more modern architectures / datasets (e.g. ResNet18 on Tiny-Imagenet), to verify if this is indeed the case.
> >
> > Regarding VIB / BNN interpretation; eq. 10 is a bit weird for a Bayesian Neural Network. The approximate posterior for the weights of a BNN should depend on the entire dataset rather than on an individual point. To me it seems that you are optimizing for something different when you let it be dependent on a single datapoint, as it seems like an i.i.d. rather than an exchangeable model. Furthermore, it seems that the uncertainty for a 10 sample CDN is worse than a single sample CDN / VIB, which is a bit puzzling.

---

> > > ### Author Response · Authors · 2018-12-03
> > > **Response to AnonReviewer1**
> > >
> > > Thank you very much for your response!
> > >
> > > Regarding scalability: Although time is to limited to verify this experimentally right now, we would like to hypothesize about the scalability of CDNs to modern DNNs. We can incorporate the more efficient parametrization proposed by Ha et al., 2017 [1] for convolutional layer, instead of using our current parametrization (Appendix D). Ha et al., 2017 [1] shows that this parametrization of (deterministic) hypernetworks requires less parameters compared to the corresponding vanilla CNN and minibatching does not seem to be an issue. Our sampling process adds some overhead over the deterministic hypernetworks, though we believe this should not be an issue either as it is simply a single matrix addition and two matrix multiplication (for the reparametrization trick).
> > >
> > > Regarding eq. 10: We agree that is super unusual and a bit weird to have an input dependent approximate posterior in an Bayesian setting. (This is why we hesitated to include the “Bayesian perspective” in the first version of our manuscript, before we were asked to include it by AnonReviewer2). We will make this more clear in the camera ready version in the case of acceptance. It is more fitting to see eq.10 from the VIB point of view, i.e. it is not an ELBO, but a lower bound of information bottleneck objective.
> > >
> > > Regarding the result for the 10 sample CDN: Indeed it is a bit worse than 1 sample CDN in OOD experiment due to the imprecision of the selection process of \lambda that we employ (as it is just a heuristic). We found (in hindsight) that other values of \lambda yield better result than the 1 sample CDN on the test set. However, we observe that the heuristic works well in general, and that the choice of specific lambda is not that crucial, e.g. in the OOD experiment, we found that training a CDN with different \lambda values between 1e-5 and 1e-7 resulted in comparable validation accuracies while leading to better uncertainty estimates compared to the baselines.
> > >
> > > References:
> > > [1] Ha et al., “Hypernetworks”, ICLR 2017.

---

### Official Review · AnonReviewer3 · 2018-11-02
**Valid idea, but not properly evaluated. Presentation of the methodology also needs work and justification of the selected distributions requires improvement.**

**Rating:** 4
**Confidence:** 4

**Review:**

In this work the authors propose an extension of mixture density networks to the continuous domain, named compound density networks. Specifically the paper builds on top of the idea of the ensemble neural networks (NNs) and introduces a stochastic neural network for handling the mixing components. The mixing distribution is also parameterised by a neural network. The authors claim that the proposed model can result in better uncertainty estimates and the experiments attempt to demonstrate the benefits of the approach, especially in cases of having to deal with adversarial attacks.

The paper in general is well written and easy to follow. I have some concerns regarding the presentation of the main objective and the lack of justification in certain parts of the methodology. Let me elaborate. First of all, I don’t understand how the main equation of the compound density network in Equation (3) is different from the general case of a Bayesian neural network? Can the authors please comment on that?

I also find weird the way that the authors arrive to their final objective in Equation (5). They start from Equation (4) which is incorrectly denoted as the log-marginal distribution while it is the same conditional distribution introduced in Equation (3) with the extra summation for all the available data points. Then they continue to Equation (5) which they present as the combination of the true likelihood with a KL regularisation term. However, what the authors implicitly did was to perform variational inference for maximising their likelihood by introducing a variational distribution q(\theta) = p(\theta | g(x_n; \psi). Is there a reason why the authors do not introduce their objective by following the variational framework?

Furthermore, in the beginning of Section 3.1 the authors present their idea on probabilistic hypernetoworks which “maps x to a distribution over parameters instead of specific value \theta.” How is this different from the case that we were considering so far? If we had a point estimate for \theta we would not require to take an expectation in Equation (3) in the first place.

My biggest concern in the methodology, however, has to do with the selection of the matrix variate normal prior for the weights and the imposition of diagonal covariances (diag(a) and diag(b)). The Kronecker product between two diagonal matrices results in another diagonal matrix, i.e., diagonal covariance, which implies that the weights within a layer are given by an independent multivariate Gaussian. What is the purpose then for introducing the matrix variate Gaussian? I would expect that you would like to impose additional structure to the weights. I expect the authors to comment on that.

Regarding the experimental evaluation of the model rather confusing. The authors have proposed a model that due to the mixing is better suited for predictions with heteroscedastic noise and can better quantify the aleatoric uncertainty. However, the selected experiments on the cubic regression toy data (Section 5.1) and the out-of-distribution classification (Section 5.2) are clear examples of system’s noise, i.e. epistemic uncertainty. The generative process of the toy data clearly states that there is no heteroscedastic noise to handle. The same applies for the notMNIST data which belong to a completely different data set compared to MNIST and thus out of sample prediction cannot benefit from the mixing; i.e., variations have to be explained by system’s noise. So overall I have the feeling that the authors have not succeeded to evaluate the model’s power with these two experiments and we cannot draw any strong conclusions regarding the benefit of the proposed mixing approach.

To continue with the experimental evaluation, I found the plots with the predictive uncertainty in Figure 3 a bit confusing. The plot by itself, as I understood, quantifies the model’s uncertainty in in- and out-of sample prediction. While I agree with the authors that it is generally desirable for a model to be more confident when predicting in MNIST (since it has already seen samples of it) compared to when predicting in notMNIST (completely different data), these plots tells us nothing regarding the predictive power of the model. There is no value in being very confident if you are wrong and vice-versa, so unless there is an accompanying plot/table reporting the accuracy I see not much value from this plot alone.

Finally, it is unclear how the authors have picked the best \lambda parameter for their approach? On page 5 they state that they “pick the value that results in a good trade-off between high uncertainty estimates and high prediction accuracy.” Does this mean that you get to observe the performance in the test in order to select the appropriate value for \lambda? If this is the case this is completely undesirable and is considered a bad practice.

---

> ### Author Response · Authors · 2018-11-23
> **Response to AnonReviewer3**
>
> We thank the reviewer for the valuable feedback! We reply to the answers and comments in the order they were raised. Note, that regarding the methodology there were some misunderstandings (which we try to avoid for future readers in the revised version).
>
> (1) The equations are indeed very related. Note however, that In a standard Bayesian neural network (BNN), one would assume that \theta is a global random variable (i.e. does not depend on input x), whereas in the CDN, we assume that \theta depends on x and is thus a local random variable. Furthermore, in a Bayesian setting p(theta|...) would play the role of a approximate posterior, which would require variational inference (VI), and thus a different objective,  to estimate it.
>
> (2) In Equation (4) we followed with  p(D | \psi) a standard notation for \sum_n p(y_n | x_n; \psi) (i.e. summation of Equation (3) wrt all data in D) which also can be found e.g. in the work of Graves (2011) [4] and Blundell et al. (2015) [5]. The objective we introduce for CDNs differs from the ELBO-based objective in VI in the way the logarithm is placed in the first term of the objective: in the ELBO we have a logarithm inside the expectation, while the logarithm is outside the expectation in the CDN objective (note however, that the sample-based approximations get equivalent if only one sample is used). Furthermore, in the ELBO we have a fixed value of \lambda = 1. We added a new Section 4 in the revised version of the paper discussing these differences. Moreover, we investigated the impact of the different objectives empirically and found that the CDN-based objective led to significantly better results, as shown in the newly added Section 6.4 in the revised manuscript.
>
> (3) Indeed we need the probabilistic version of hypernetworks to implement the model we described in Equation (3).  We just wanted to point out that this is in contrast to the vanilla  hypernetworks proposed by Ha et al. (2016) [1] and Jia et al.  (2016) [2] which would produce a point estimate for \theta.
>
> (4) We used a matrix-variate normal (MVN) to reduce the parameters of the model. Using a diagonal MVN for X \in R^{p x q} one needs pq+p+q parameters. In contrast a fully-factorized diagonal Gaussian needs pq+pq. But you are right, we could easily extend our approach to account for more flexible distributions by using a "diagonal plus rank-one" structure diag(a)+uu^T, with vectors a and u, as noted by Louizos and Welling ( 2016) [3] (the increase of parameters is negligible: adding additional vector u). We will investigate the benefits of more flexible mixing distributions in future work.
>
> (5) Thanks for this valuable comment! We have revised the toy experiments to include a heteroscedastic regression task (see Section 6.1.). It shows that CDNs are able to quantify the heteroscedastic aleatoric uncertainty. However, we kept the OOD experiments on MNIST and notMNIST in the paper as well, since we consider it as very interesting that the CDN, while being designed for modelling aleatoric uncertainty, is very competitive on this task. Moreover, we investigate the mixing distribution learned in Appendix G.
>
> (6) Of course, you are right! Previously we showed the test accuracy in Appendix F.  To make it more directly accessible, we have now added the test accuracy achieved by the different models into the legends of the plots, showing that CDN achieves similar predictive power as the baselines. We now present the validation accuracy instead in Appendix F.
>
> (7) Sorry, for this unfortunate formulation! We observed that generally as \lambda increases, the uncertainty is increasing, while the accuracy is decreasing. Therefore a simple and effective heuristic for choosing \lambda is to look at the validation set of MNIST and choose the highest \lambda that still results in high accuracy (e.g. >. 0.97). We have made this procedure clear in the revised manuscript.
>
>
> References:
> [1] Ha, David, Andrew Dai, and Quoc V. Le. "Hypernetworks." arXiv preprint arXiv:1609.09106 (2016).
> [2] Jia, Xu, et al. "Dynamic filter networks." Advances in Neural Information Processing Systems. 2016.
> [3] Louizos, Christos, and Max Welling. "Structured and efficient variational deep learning with matrix gaussian posteriors." International Conference on Machine Learning. 2016.
> [4] Graves, A. (2011). Practical variational inference for neural networks. In Advances in neural information processing systems (pp. 2348-2356).
> [5] Blundell, C., Cornebise, J., Kavukcuoglu, K. & Wierstra, D.. (2015). Weight Uncertainty in Neural Network. Proceedings of the 32nd International Conference on Machine Learning, in PMLR 37:1613-1622

---

> > ### Comment · AnonReviewer3 · 2018-11-27
> > **Much better version but I still have my reservations.**
> >
> > I would like to thank the authors for their response and their effort put in improving the manuscript based on the reviewer's comments.
> >
> > Regarding equation 4: I admit it was my fault, I did not see the place of the logarithm.
> >
> > Regarding the MVN prior: I understand that the diagonal structure leads to less parameters, however, I still find it undesirable the imposition of such a hard structure on the parameters of the network (by construction, all rows and columns of W_l are constrained to be independent).
> >
> > Regarding aleatoric uncertainty: Thank you for including the additional experiment with the heteroscedastic regression; it definitely improved the evaluation while also demonstrates the benefit of the proposed approach. On the contrary I keep wondering what is the purpose of the OOD experiment. It surely demonstrates some nice results, however, we are left to wonder why does the proposed approach perform better. Is there any benefit from the mixture component and why? Can the authors please comment on that?
> >
> > Finally, I am still negative regarding the selection of the \lambda parameter and I am not convinced by the authors response. In a real problem there is no access to the test set. Of course one can select a part of the training set as a validation set, but from what I understood this is not the setup followed in the current experimental procedure.
> >
> > Overall, I appreciate the authors' effort, and based on their response and their additional comparisons I am willing to up my score but I am still not sure that it is good enough for publication.

---

> > > ### Author Response · Authors · 2018-11-30
> > > **Response to AnonReviewer3**
> > >
> > > We thank the reviewer very much for the follow-up!
> > >
> > > Regarding the MVN distribution: While we agree that indeed it imposes a hard structure, it works very well in practice in related methods, e.g. those of Louizos and Welling, 2016 [1] and Blundell et al., 2015 [2], which motivated us to use it in our model as well, and makes us believe this should not result in a downrating of our paper.
> > >
> > > Regarding the OOD experiment: We originally used the OOD experiment to gain insight on how our approach compares with BNNs. We also found that the result is surprising. A recent work by Alemi et al., 2018 [3] shows that the VIB  model they analyze is also able to quantify uncertainty of OOD data. They explain the behavior by arguing that VIB is better calibrated than standard NN and that the prior in the KL-term corresponds to density estimation in the latent space (note that they learn the prior p(z), although they fixed it to N(0, I) in the original paper [Alemi et al., 2017]). This arguments naturally carry over to CDNs, due to the connection to VIB (for one sample approximations).
> > >
> > > Regarding the selection of \lambda:  choosing \lambda based on the **validation set** (not the test set!) is exactly the selection heuristic we follow in all experiments presented in the revised version! We are sorry, that our formulations where still not precise enough.
> > >
> > > References:
> > > [1] Louizos, Christos, and Max Welling. "Structured and efficient variational deep learning with matrix gaussian posteriors." International Conference on Machine Learning. 2016.
> > > [2] Blundell, C., Cornebise, J., Kavukcuoglu, K. & Wierstra, D.. (2015). Weight Uncertainty in Neural Network. Proceedings of the 32nd International Conference on Machine Learning, in PMLR 37:1613-1622
> > > [3] Alemi, Alexander A., Ian Fischer, and Joshua V. Dillon. "Uncertainty in the variational information bottleneck." arXiv preprint arXiv:1807.00906 (2018).
> > > [4] Alemi, Alexander A., et al. "Deep variational information bottleneck." ICLR 2017.

---

### Author Response · Authors · 2018-11-23
**General response to all of the reviewers and manuscript changelog**

We thank the reviewers for the very valuable feedback and suggestions. We are grateful and believe they lead to a crucial improvement of our paper, for which we uploaded a revised version. The  changes are summarized in the following:

We, most importantly, have
1) added a description of the connections between the CDN objective, and those used for variational inference for Bayesian neural networks and variational information bottleneck (Sec. 4),
2) investigated the difference between these objectives experimentally, finding that the CDN objective performs best (Sec. 6.4),
3) reworked the toy regression experiment (Sec. 6.1) to better reflect aleatoric uncertainty, and
4) rerun the baseline models in the experiments with the hyperparameters suggested in the respective original publications (summarized in Appendix F).

Moreover, we have
1) made more clear that we are using a diagonal matrix normal distribution as the mixing distribution, instead of a fully-factorized Gaussian, to gain parameter efficiency,
2) added the corresponding test accuracy (which was previously only shown in the appendix) to the legend of the OOD experiment’s figures,
3) clarified the hyperparameter (\lambda) selection strategy we used in the experiments, and
4) fixed an isolated bug found in the toy regression experiment.

---

### Meta-Review · Area_Chair1 · 2018-12-14

**Confidence:** 2
**Recommendation:** Reject

**Metareview:**

Reviewers are in a consensus and weakly recommended to reject after engaging with the authors, with the reviewers updating their scores on Dec 11 after engagement. The authors answered most of the reviewers' concerns, however from further discussions with the  reviewers there are still some points which lead them to rank the paper lower than others. I thus lean to reject. Please take reviewers' comments into consideration to improve submission should you choose to resubmit.